

# Value of preoperative $^{18}$F-FDG PET/CT and HRCT in predicting the differentiation degree of lung adenocarcinoma dominated by solid density

Xiaolin Chen[1,*], Ping Li[1,*], Minghui Zhang[2], Xuewei Wang[1] and Dalong Wang[1]

[1] Department of PET/CT, The Second Affiliated Hospital of Harbin Medical University, Harbin, Heilongjiang Province, China

[2] Department of Medical Oncology, Harbin Medical University Cancer Hospital, Harbin, China

[*] These authors contributed equally to this work.

Corresponding author
Dalong Wang,
wangdalongbao@163.com

## ABSTRACT

**Purpose**. To evaluate the value of positron emission tomography/computed tomography (PET/CT) combined with high-resolution CT (HRCT) in determining the degree of differentiation of lung adenocarcinoma.

**Methods**. From January 2018 to January 2022, 88 patients with solid density nodules that are lung adenocarcinoma were surgically treated. All patients were examined using HRCT and PET/CT before surgery. During HRCT, two independent observers assessed the presence of lobulation, spiculation, pleural indentation, vascular convergence, and air bronchial signs (bronchial distortion and bronchial disruption). The diameter and CT value of the nodules were measured simultaneously. During PET/CT, the maximum standard uptake value (SUVmax), mean standard uptake value (SUVmean), metabolic tumor volume (MTV), and total lesion glycolysis (TLG) of the nodules were measured. The risk factors of pathological classification were predicted by logistic regression analysis.

**Results**. All 88 patients (mean age $60 \pm 8$ years; 44 males and 44 females) were evaluated. The average nodule size was $2.6 \pm 1.1$ cm. The univariate analysis showed that carcinoembryonic antigen (CEA), pleural indentation, vascular convergence, bronchial distortion, and higher SUVmax were more common in poor differentiated lung adenocarcinoma, and in the multivariate analysis, pleural indentation, vascular convergence, and SUVmax were predictive factors. The combined diagnosis using these three factors showed that the area under the curve (AUC) was 0.735.

**Conclusion**. SUVmax >6.99 combined with HRCT (pleural indentation sign and vascular convergence sign) is helpful to predict the differentiation degree of lung adenocarcinoma dominated by solid density.

## INTRODUCTION

In 2018, lung cancer surpassed breast, prostate, and colorectal cancer, becoming the leading cause of cancer death in the United States (*Siegel et al., 2021*). As a type of

lung adenocarcinoma, invasive adenocarcinoma can be divided into lepidic (LPA), acinar (APA), papillary (PPA), micropapillary (MPA), and solid (SPA) predominant adenocarcinoma according to the proportion of the main components, when two pathological components are found, classification is based on the larger component (*Travis et al., 2015*). Furthermore, *Kim et al. (2019)* and *Russell et al. (2011)* divided lung adenocarcinoma into three groups according to the postoperative prognosis of lung adenocarcinoma: low (adenocarcinoma *in situ* (AIS) + minimally invasive adenocarcinoma (MIA) + lepidic predominant adenocarcinoma (LPA)), intermediate (acinar (APA) + papillary predominant adenocarcinoma (PPA)), and high (micropapillary (MPA) + solid pulmonary adenocarcinoma (SPA)) risk.

$^{18}$F-fluorodeoxyglucose (FDG) positron emission tomography/computed tomography (PET/CT) is important in the staging, detection, and treatment of tumors (*Fonti, Conson & Vecchio, 2019*). The widely used metabolism parameter for diagnosing malignant tumor was the maximum standard uptake value (SUVmax) (*Fonti, Conson & Vecchio, 2019*). Compared with conventional SUVmax, volume parameters such as metabolic tumor volume (MTV) and total lesion glycolysis (TLG) can evaluate the metabolic activity of the whole tumor mass (*Tosi et al., 2021*). MTV is a semiquantitative parameter that integrates metabolism and volume (*Chen et al., 2017*), whereas TLG is the product of lesion volume and standard uptake value (SUV). Both are useful indicators of tumor load and invasiveness, and are important in the prognosis of tumors (*Tosi et al., 2021*). However, few articles have studied the relationship between these parameters and the pathological types of lung adenocarcinoma. Among lung adenocarcinoma, the metabolic parameter SUVmax of invasive lung adenocarcinoma (except for lepidic type) was significantly higher than that of other types (*Suárez-Piñera et al., 2018*), the prognoses of the various pathological types differ (*Cha et al., 2014*). Hence, it is vital to study the heterogeneity of invasive lung adenocarcinoma. However, there are few studies about the relationship between PET parameters and the pathological types dominated by solid density in lung adenocarcinoma.

Under high-resolution CT (HRCT), lung adenocarcinoma can be classified into pure ground glass nodules, mixed ground glass nodules, and solid nodules (*Wu et al., 2020*). Current studies (*Kudo et al., 2015*) have shown that lung adenocarcinomas in highly differentiated groups are usually characterized by ground glass nodules, whereas solid density nodules are generally observed among the invasive pathological types. According to lung adenocarcinoma with ground glass nodules, previous studies have found that with increasing CT value (HU) and large size, the possible types of invasive lung adenocarcinoma also increase, and the SUVmax helped distinguish different pathological types and degrees of differentiation (*Yanagawa et al., 2020*; *Sun et al., 2021*; *Li et al., 2020*). By contrast, *Hu et al. (2017)* studied 54 cases of ground glass nodules and found that when the diameter of ground glass nodules were <1 cm, no obvious metabolism can be observed in PET/CT for pure or mixed ground glass nodules, even they are the malignant pathological type. In PET/CT, the ground glass composition in mixed ground glass nodules may not be significantly absorbed and their solid parts are always too small for PET resolution. Therefore, it is more reasonable to select solid density nodules (no visible ground-glass component on thin-section CT) and
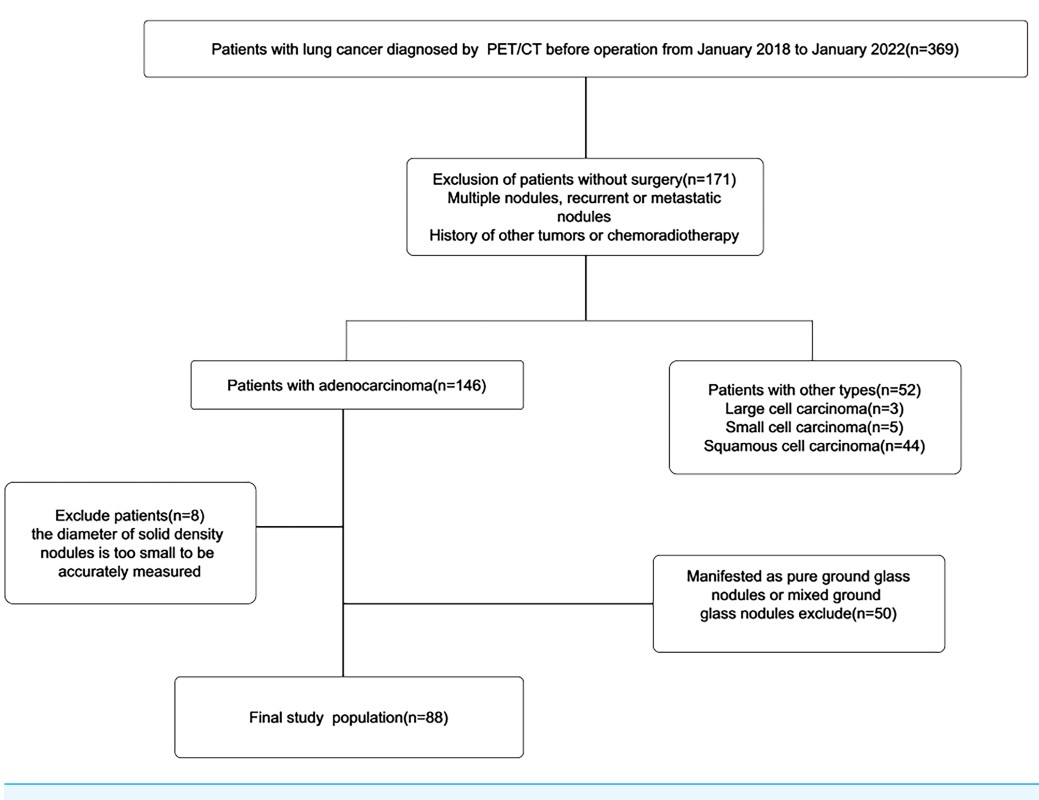

**Figure 1  Flowchart of the patient.**

exclude pure and mixed ground glass nodules when studying the correlation between FDG metabolism and tumor differentiation in lung adenocarcinoma. We hypothesized that the tumor differentiation of lung adenocarcinomas presenting as solid nodules might be better characterized *via* HRCT and PET/CT. Therefore, our research aimed to use PET/CT and HRCT to predict the differentiation degree of lung adenocarcinoma dominated by solid density.

# MATERIAL AND METHODS

## Patient characteristics

The Harbin Medical University Institutional Human Ethics Review Board approved this retrospective study (No. HMUIRB20160008), and the requirement for informed consent was waived. A study had been conducted on patients who underwent lung cancer surgery in the Second Affiliated Hospital of the Harbin Medical University from January 2018 to January 2022. Overall, 369 patients underwent HRCT and PET/CT before surgery (within 4 weeks). The exclusion criteria were: (1) Multiple nodules, recurrent or metastatic nodules, (2) history of other tumors or chemoradiotherapy, (3) other pathological types of lung cancer, (4) ground glass and mixed ground glass nodules on HRCT, (5) nodules with diameters that are too small (<1 cm) to be accurately measured, and (6) unclear pathological types. The final study contains a total number of 88 patients (Fig. 1).

## Pathological subtype and histological grade

Lung adenocarcinoma was divided into three groups according to the postoperative prognosis of lung adenocarcinoma: well differentiation (adenocarcinoma *in situ* (AIS) + minimally invasive adenocarcinoma (MIA) + lepidic predominant adenocarcinoma (LPA)), moderate differentiation (acinar (APA) + papillary predominant adenocarcinoma(PPA)) and poor differentiation (micropapillary (MPA) + solid pulmonary adenocarcinoma(SPA)) (*Kim et al., 2019*).

## $^{18}$F-FDG PET/CT

All patients fasted for 6 h prior to the procedure and kept their blood glucose levels <7 mmol/L before undergoing $^{18}$F-FDG PET/CT (Discovery ST, GE Medical Systems, Milwaukee, WI). When we finished the intravenous injection of 3.7 MBq/kg of FDG and waited for 60 min, we scanned patients from the skull base to the upper femur. The CT scanning parameters were set as follows: tube current, 35 mA, tube voltage, 120 KV, rotation time, 1.5−1.8 s, selected range, same as that for PET, collimation, 0.6 mm or 1.2 mm, scanning layer thickness, 1.0 mm, layer spacing, 1.0 mm, and bed speed, generated automatically. The patients laid on the examination bed on their back and their posture was kept unchanged. The captured CT and PET images were transmitted to the SyngoTRUED workstation equipped by Siemens PET/CT for fusion. If necessary, 3D reconstruction of the image was performed.

## MTV and TLG measurements

In this study, PET/CT imaging was evaluated by two experienced doctors. Disagreements were resolved through discussion until a consensus was reached. MTV was measured from FDG PET/CT images by an automatic contour rendering program based on SUVmax. The edge of the tumor was defined according to the threshold of 40% SUVmax in the volume of interest in the contour edge. The TLG was calculated by multiplying the MTV of each lesion with the corresponding average SUV determined in a selected contouring volume of interest.

## Statistical analysis

All of the data in this study were analyzed and processed using SPSS 20.0 (SPSS Inc, Chicago, IL). Data under normal distribution are expressed as mean ± standard deviation ($\bar{x}$ ± s); otherwise, they are expressed as median (interquartile range). The categorical data are expressed as number (%). The differences of lesion size, density, SUVmax, SUVmean, MTV and TLG between the two groups were tested by t test, whereas the differences of lobulation sign, spiculation sign, air bronchiologram, vascular convergence, pleural indentation and pleural contact between the two groups were analyzed by chi-square test. Univariate logistic regression analysis was used to evaluate the relationship between poor and moderate differentiation and HRCT signs, clinical parameters, and PET/CT parameters. Meaningful parameters were included in the multivariate analysis (stepwise binary logistic regression analysis). The diagnostic values of different parameters were compared by the receiver operating characteristic curve (ROC) and the area under the curve (AUC), the sensitivity and specificity were calculated, and the best cut-off value for

**Table 1  Characteristics of 88 patients.**

| Patient characteristics | Moderate differentiation (60) | Poor differentiation (28) |
|---|---|---|
| Age (Year) | $60.2 \pm 8.1$ | $60.4 \pm 9.1$ |
| Lobe location | | |
| Right upper lobe N (%) | 17(28.3) | 9(32.1) |
| Right middle lobe N (%) | 2(3.3) | 4(14.3) |
| Right lower lobe N (%) | 14(23.4) | 4(14.3) |
| Left upper lobe N (%) | 19(31.7) | 7(25.0) |
| Left lower lobe N (%) | 8(13.3) | 4(14.3) |
| Gender | | |
| Male N (%) | 28(46.7) | 16(57.1) |
| Female N (%) | 32(53.3) | 12(42.9) |
| Smoking | | |
| Yes N (%) | 19(31.7) | 17(60.7) |
| No N (%) | 41(68.3) | 11(39.3) |
| CEA | | |
| >5ng/ml N (%) | 19(31.7) | 15(53.6) |
| <5ng/ml N (%) | 41(68.3) | 13(46.4) |
| Lymph node metastasis | | |
| Yes N (%) | 12(20.0) | 11(39.3) |
| No N (%) | 48(80.0) | 17(60.7) |
| Clinical T descriptor | | |
| T1 | | |
| T1a N (%) | 1(1.7) | 0 |
| T1b N (%) | 17(28.3) | 7(25.0) |
| T1c N (%) | 30(50.0) | 12(42.9) |
| T2 | | |
| T2a N (%) | 7(11.7) | 5(17.8) |
| T2b N (%) | 5(8.3) | 4(14.3) |

**Notes.**
Note—Age is shown as means ± standard deviations. The remaining data are numbers of patients.
CEA, carcinoembryonic antigen.

the categorization of low and high SUVmax was determined by maximizing the Youden index. A $p$-value <0.05 was considered statistically significant.

# RESULTS

## Patient characteristics

A total of 88 patients with lung adenocarcinoma were included, including 44 males and 44 females, with an average age of 60 years (range: 37–78 years). The breakdown for the pathological types were: 48 AP, 12 PP, 17 SP, and 11 MP. Moderate differentiation (60) and poor differentiation (28) were founded. The main information of 88 patients is shown in Table 1.

**Table 2  HRCT and PET/CT manifestations of different differentiations.**

| Tumor Characteristic | Moderate differentiation (60) | | Poor differentiation (28) | | P Value |
|---|---|---|---|---|---|
| | Acinar (48) | Papillary (12) | Micropapillary (11) | Solid (17) | |
| Nodule diameter (cm) | 2.6 ± 1.3 | 2.2 ± 0.7 | 2.6 ± 0.6 | 2.8 ± 1.4 | 0.308 |
| CT value (HU) | 27.9 ± 9.2 | 28.7 ± 9.4 | 31.0 ± 7.0 | 31.3 ± 5.9 | 0.122 |
| Radiological characteristics | | | | | |
| Lobulation N (%) | 41(85.4) | 10(83.3) | 7(63.4) | 16(94.1) | 0.977 |
| Spiculation N (%) | 29(60.4) | 9(75.0) | 9(81.8) | 13(76.5) | 0.153 |
| Vascular convergence N (%) | 29(60.4) | 5(41.7) | 11(100.0) | 12(70.6) | 0.020 |
| Pleural contact N (%) | 36(75.0) | 12(100.0) | 9(81.8) | 14(82.4) | 0.813 |
| Pleural indentation N (%) | 9(18.8) | 2(16.7) | 5(45.4) | 7(41.2) | 0.015 |
| Air bronchiologram | | | | | |
| Dilatation N (%) | 14(29.2) | 5(41.7) | 5(45.4) | 8(47.1) | 0.180 |
| Disruption N (%) | 3(6.3) | 1(8.3) | 4(36.4) | 0 | 0.447 |
| Distortion N (%) | 9(18.8) | 2(16.7) | 4(36.4) | 7(41.2) | 0.034 |
| SUVmax | 8.2 ± 5.2 | 5.3 ± 3.1 | 8.2 ± 3.1 | 10.7 ± 4.6 | 0.021 |
| SUVmean | 4.4 ± 3.2 | 2.5 ± 1.4 | 4.2 ± 2.1 | 5.9 ± 2.7 | 0.050 |
| MTV | 4.8 ± 6.2 | 2.1 ± 1.5 | 5.2 ± 3.9 | 8.2 ± 14.2 | 0.119 |
| TLG | 31.3 ± 47.5 | 8.6 ± 7.1 | 26.7 ± 20.8 | 65.3 ± 107.4 | 0.082 |

Notes.

SUVmax, maximum standard uptake value; SUVmean, mean standard uptake value; MTV, metabolic tumor volume; TLG, total lesion glycolysis.

$P$ value is for comparison of moderate differentiation (acinar and papillary) and poor differentiation (micropapillary and solid).

The measurement data between the two groups were tested by $t$ test.

The counting data between the two groups were tested by chi-square.

## The manifestation of lung adenocarcinoma with different differentiations under HRCT and PET/CT

Table 2 summarizes the manifestations of the different differentiations of lung adenocarcinomas under HRCT and PET/CT. In HRCT, moderate and poor differentiated lung adenocarcinomas differed in terms of vascular convergence ($p = 0.020$), pleural indentation ($p = 0.015$), and bronchial distortion ($p = 0.034$). However, no significant difference was observed in terms of lobulation ($p = 0.977$), spiculation ($p = 0.153$), bronchial disruption ($p = 0.447$), and dilatation ($p = 0.180$). Similarly, there was no difference in nodule size ($p = 0.308$) and CT value ($p = 0.122$) between the different differentiations. In PET/CT, the SUVmax ($p = 0.021$) of poor differentiated lung adenocarcinoma were significantly higher than those of moderate differentiation (Fig. 2). However, the MTV ($p = 0.119$), SUVmean ($p = 0.050$), and TLG ($p = 0.082$) had no significant difference between the two groups. The SUVmax of the different pathological subtypes: the acinar, papillary, micropapillary, and solid types had values of 8.2 ± 5,2, 5.3 ± 3.1, 8.2 ± 3.1, and 10.7 ± 4.6, respectively. The SUVmax had statistically significant differences between the APA and PPA in moderated group( $p = 0.043$), however, no such statistical significance was found between MPA and SPA in poor differentiated group (Fig. 3). The selected patients of different pathological subtypes were shown in Fig. 4.
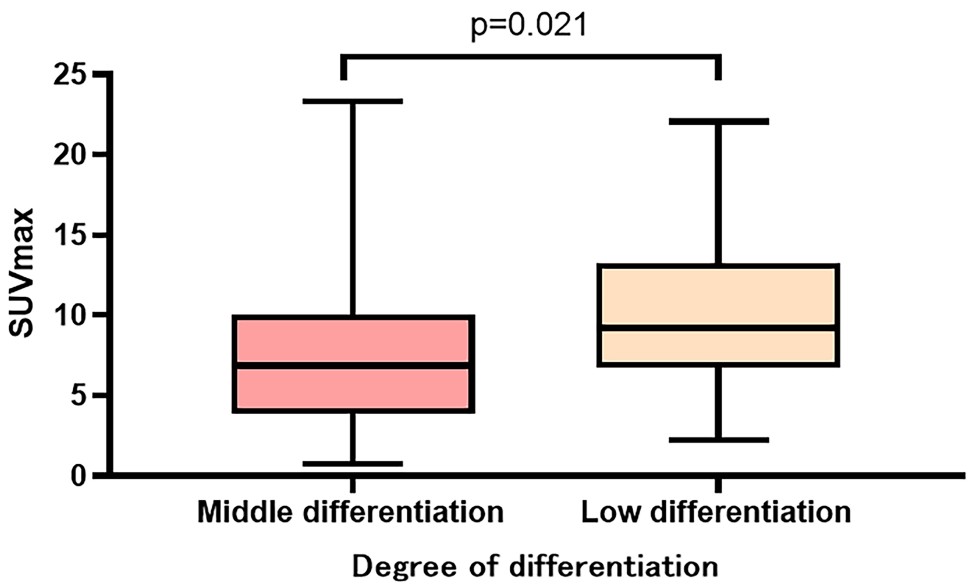

**Figure 2**  SUVmax in different degrees of differentiation.

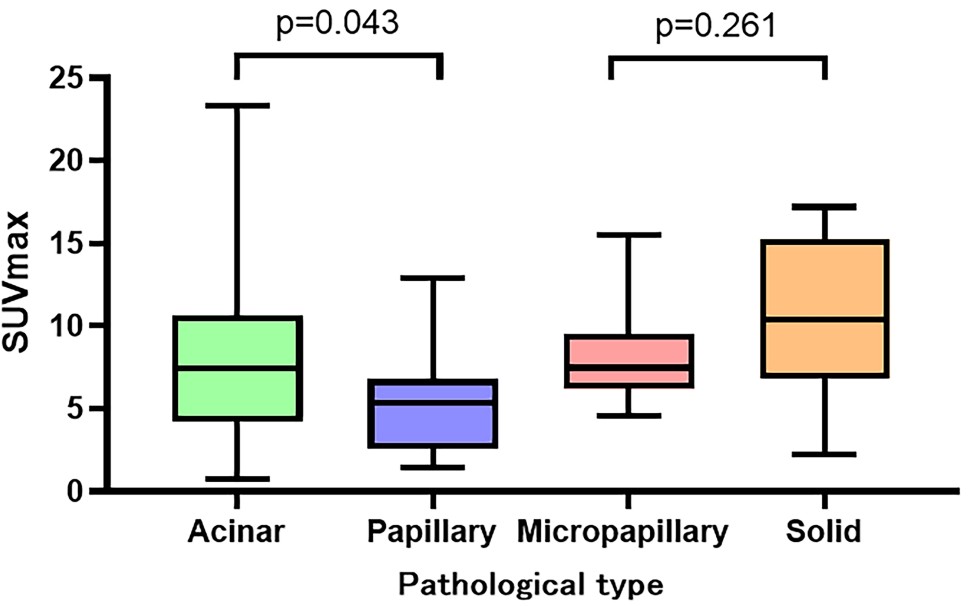

**Figure 3**  SUVmax in different pathological subtypes.

## Univariate and multivariate analyses of lung adenocarcinoma differentiation

Table 3 summarizes the results of HRCT findings (lobulation, spiculation, pleural indentation, pleural contact, vascular convergence, air bronchial signs (distortion, disruption, and dilatation), solid nodule size, and CT value of solid nodules), CEA, and

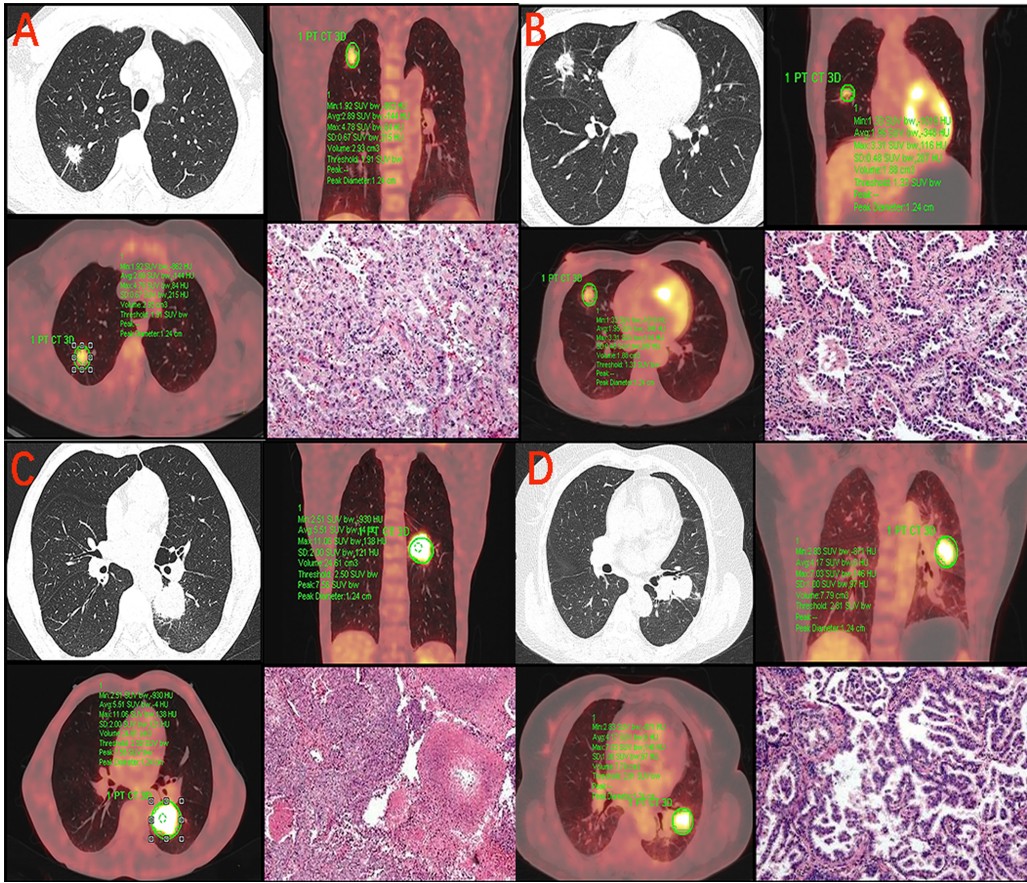

**Figure 4** **SUVmax in the different pathological subtypes of lung adenocarcinoma.** (A) 49-year-old male with adenocarcinoma in acinar predominant. CT image shows a 15 mm solid nodule in the right upper lobe. PET/CT fusion image shows FDG uptake with SUVmax of 4.78 and MTV of 2.93 cm³. Hematoxylin-eosin staining (HE). (B) 43-year-old female with adenocarcinoma in papillary predominant. CT image shows a 18 mm solid nodule in the right middle lobe. PET/CT fusion image shows FDG uptake with SU-Vmax of 3.31 and MTV of 1.88 cm³ (HE). (C) 60-year-old male with adenocarcinoma in solid predominant. CT image shows a 35 mm solid nodule in the left lower lobe. PET/CT fusion image shows FDG uptake with SUVmax of 11.06 and MTV of 24.61 cm³ (HE). (D) 64-year-old female with adenocarcinoma in micropapillary predominant. CT image shows a 26 mm solid nodule in the left upper lobe. PET/CT fusion image shows FDG uptake with SUVmax of 7.03 and MTV of 7.79 cm³ (HE).

PET/CT parameters (SUVmax, SUVmean, and TLG). In the univariate logistic regression analysis, pleural indentation, vascular convergence, distortion of air bronchiologram, CEA , and SUVmax predicted the invasiveness of lung adenocarcinoma ($p < 0.05$). These variables with $p < 0.05$ in the univariate analysis were then incorporated into the multivariate analysis. In the multivariate analysis (Table 4), pleural indentation (Odds ratio = 5.453,95% confidence interval (CI), [1.349–15.570], $p = 0.034$), vascular convergence (OR = 6.126,95% CI [1.788–25.150], $p = 0.020$), and SUVmax (OR = 1.785,95% CI [1.524–1.969]; $p = 0.031$) could predict the degree of malignancy. The cut-off of SUVmax for moderate differentiated and poor differentiated lung adenocarcinoma was 6.99 (sensitivity, 75%, specificity, 53.3%). The diagnostic efficiency of solid density lung adenocarcinoma

**Table 3  Univariable analysis of lung adenocarcinoma with different differentiation.**

| | Univariable Analysis OR | 95%CI | *P* value |
|---|---|---|---|
| Nodule diameter (cm) | 1.019 | 0.982–1.058 | 0.314 |
| CT value (HU) | 1.045 | 0.988–1.105 | 0.125 |
| Pleural indentation N (%) | 3.341 | 1.237–9.026 | 0.017 |
| Pleural contact N (%) | 1.150 | 0.362–3.652 | 0.813 |
| Lobulation N (%) | 0.812 | 0.245–2.692 | 0.733 |
| Spiculation N (%) | 2.123 | 0.747–6.032 | 0.158 |
| Vascular convergence N (%) | 3.518 | 1.178–10.501 | 0.024 |
| Air bronchiologram | | | |
| Disruption N (%) | 2.333 | 0.593–10.109 | 0.257 |
| Distortion N (%) | 2.882 | 1.059–7.846 | 0.038 |
| Dilation N (%) | 1.870 | 0.745–4.696 | 0.183 |
| CEA N (%) | 3.111 | 1.227–7.886 | 0.017 |
| SUVmax | 1.116 | 1.013–1.229 | 0.026 |
| SUVmean | 1.166 | 0.994–1.368 | 0.059 |
| TLG | 1.007 | 0.998–1.015 | 0.125 |

**Notes.**
CEA, carcinoembryonic antigen; OR, Odds ratio; 95%CI, 95% confidence interval; SUVmax, maximum standard uptake value; SUVmean, mean standard uptake value; TLG, total lesion glycolysis.

**Table 4  Multivariable analysis of lung adenocarcinoma with different differentiation.**

| | Multivariable Analysis OR | 95% CI | *P* value |
|---|---|---|---|
| CEA N (%) | 1.512 | 0.129–1.682 | 0.224 |
| Pleural indentation N (%) | 5.453 | 1.349–15.570 | 0.034 |
| Vascular convergence N (%) | 6.126 | 1.788–25.150 | 0.020 |
| Distortion of air bronchiologram N (%) | 2.085 | 2.009–14.094 | 0.062 |
| SUVmax | 1.785 | 1.524–1.969 | 0.031 |

**Notes.**
CEA, carcinoembryonic antigen; OR, Odds ratio; 95%CI, 95% confidence interval; SUVmax, maximum standard uptake value.

was evaluated by ROC, and the results showed that the AUC of SUVmax combined with vascular convergence, SUVmax combined with pleural indentation, SUVmax combined with vascular convergence and pleural indentation were 0.701, 0.701, and 0.735, respectively (Fig. 5 and Table 5).

# DISCUSSION

Lung adenocarcinoma is the most common of non-small cell lung cancer (*Ercelep et al., 2021*), which can be divided into different histological types according to its heterogeneity. However, some lung adenocarcinomas with pure and mixed ground glass nodules have no FDG uptake in PET/CT (*Hu et al., 2017*), and there is little research focusing on the invasiveness of solid density lung adenocarcinoma using PET/CT. Here, we studied the

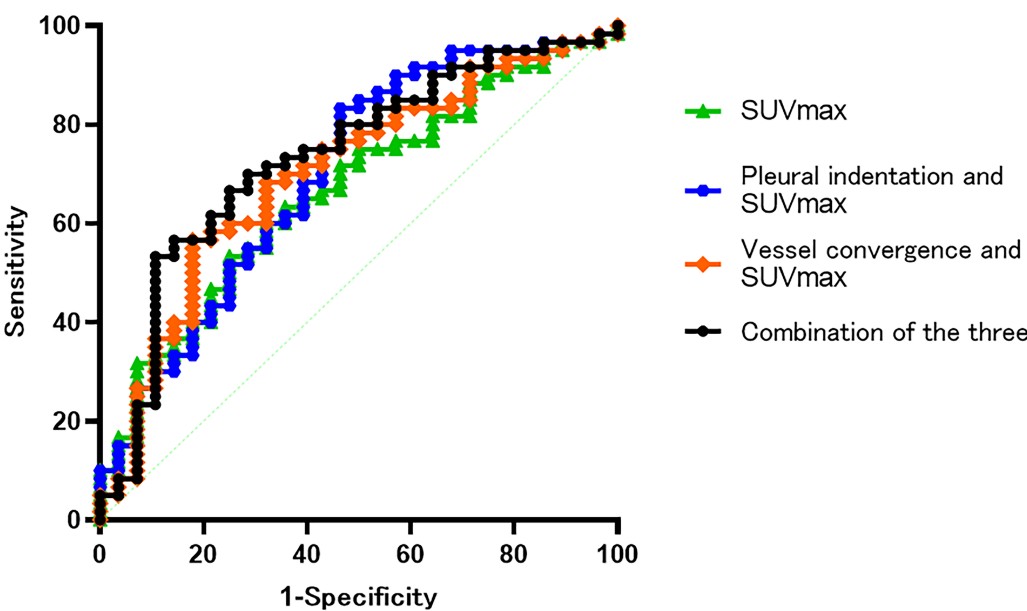

**Figure 5** The results showed that the AUC of SUVmax and vascular convergence, SUVmax and pleural indentation, SUVmax combined pleural indentation and vascular convergence were 0.701, 0.701 and 0.735, respectively.

**Table 5** Diagnostic performances of different variables in distinguishing the moderate and poor groups.

| | Cut-off value | sensitivity | specificity | AUC |
|---|---|---|---|---|
| SUVmax | 6.99 | 75.00 | 53.30 | 0.666 |
| Vascular convergence | – | – | – | 0.627 |
| Pleural indentation | | | | 0.623 |
| SUVmax and vascular convergence | – | 82.10 | 43.30 | 0.701 |
| SUVmax and pleural indentation | – | 53.60 | 83.30 | 0.701 |
| SUVmax, pleural indentation and vascular convergence | – | 89.30 | 53.30 | 0.735 |

**Notes.**

AUC, Area under curve; SUVmax, maximum standard uptake value.

manifestations of the solid density nodules of lung adenocarcinoma in HRCT and PET/CT to further distinguish the degree of differentiation in lung adenocarcinoma dominated by solid density.

In our study, all solid density nodules were invasive lung adenocarcinomas, which is consistent with previous studies (*Yanagawa et al., 2020*). The pathological types were papillary, acinar, micropapillary, and solid, but no lepidic type was observed, which is consistent with the results of *Austin et al. (2013)*.

Only moderate and poor differentiated tumors were founded in our study of surgical lung adenocarcinoma dominated by solid density. According to HRCT, the univariate analysis showed that poor differentiated lung adenocarcinoma is more likely to have CT signs such as pleural indentation ($p = 0.017$), vascular convergence ($p = 0.024$), and bronchial distortion ($p = 0.038$) compared to moderate differentiated adenocarcinoma. Similarly, in our multivariate studies, pleural indentation and vascular convergence were more likely to occur in poor differentiated than in the moderate differentiated lung adenocarcinoma. Pleural indentation is caused by the contraction of fibers in the tumor (*Seki et al., 2007*). Studies (*Li et al., 2021*) have also shown that vascular convergence reflects angiogenesis. The occurrence of bronchial distortion may also be caused by the contraction and traction of fibrosis in the tumor. Moreover, others (*Li et al., 2020*) have shown increased CT value in ground glass nodules are correlated with invasiveness of lung adenocarcinoma. However, in our study, CT value is not a risk factor for predicting invasive lung adenocarcinoma ($p = 0.125$), this could be due to the inclusion of solid density nodules.

FDG uptake is regulated by glucose metabolism. Previous studies have shown that tumor FDG uptake correlates with glucose transporter-1 in tumors (*Ichikawa et al., 2019*). However, as the metabolism of some ground glass nodules in PET/CT is low, the metabolic parameters cannot be measured accurately. Therefore, we focused on the metabolism of lung adenocarcinoma dominated by solid density under PET/CT. Based on our findings, moderate and poor differentiated solid nodules significantly differed in terms of SUVmax ($p = 0.021$). Among the four types of invasive lung adenocarcinomas, the highest SUVmax was found in solid lung adenocarcinoma, which is consistent with other studies (*Suárez-Piñera et al., 2018*; *Sun et al., 2021*). SUVmax can help distinguish the heterogeneity of invasive lung adenocarcinoma and it is very important in monitoring the prognosis of patients. Furthermore, we found that the SUVmax significantly differed ($p = 0.043$) between APA and PPA in moderate differentiated lung adenocarcinoma, but not ($p = 0.261$) between MPA and SPA in poor differentiated lung adenocarcinoma , which is similar with other studies (*Sun et al., 2021*), further study should be performed to observe the relationship between SUVmax and pathological subtypes. In our multivariate analysis, we found that SUVmax differed between the moderate and poor differentiated solid nodules. SUVmax is an independent predictor of poor differentiation, which is of clinical value in judging the invasiveness of lung adenocarcinoma. By contrast, MTV and TLG were considered to be more sensitive to tumor load and invasiveness than SUVmax in a previous study by *Carretta et al. (2020)*. However, there are few studies on these two parameters in terms of predicting invasiveness. One previous study (*Wang et al., 2017*) found that tumors with worse the pathological grades had higher MTV and TLG values. In our study, MTV and TLG were unable to predict the pathological type and differentiation of invasive lung adenocarcinoma. The relation between metabolism volume parameters (MTV and TLG) and the differentiations should be further study. We also found that the SUVmax cut-off value between moderate and poor differentiated lung adenocarcinoma was 6.99. At greater values, poor differentiation (MPA or SPA) was more likely, this is similar to the results of *Cha et al. (2014)*. SUVmax, vascular convergence sign and pleural indentation sign were independent influencing factors for identifying the poor differentiation. The

diagnostic efficacy of SUVmax combined with vascular convergence, SUVmax combined with pleural indentation, SUVmax combined with vascular convergence and pleural indentation were 0.701, 0.701 and 0.735 respectively. The sensitivity of combined diagnosis were 82.1%, 53.6% and 89.3%, respectively, and the specificity were 43.3%, 83.3%, 53.3%, respectively. Therefore, PET metabolism parameter (SUVmax>6.99) combined with the HRCT parameters (vascular convergence sign, pleural indentation sign) could improve the diagnosis of poorly differentiated adenocarcinoma.

In previous studies (*Yue et al., 2018*), increasing diameters of solid nodules increased the likelihood of invasive lung adenocarcinoma, for the solid components are related to tumor proliferation and fibrous lesions (*Gandara et al., 2006*). Furthermore, another study (*Li et al., 2021*) revealed that the diameter of nodules increased faster in lung adenocarcinoma with micropapillary and solid components than in the other types. However, our study showed that there was no correlation between the maximum diameter of the nodules and the SUVmax in lung adenocarcinoma with solid density.

In lung adenocarcinoma with mediastinal lymph node metastasis, determining the nature of the lymph nodes is crucial in selecting the appropriate treatment and evaluating prognosis (*Miao et al., 2019*). PET/CT has become an indispensable method in evaluating lung cancer staging. We found that poor differentiated lung adenocarcinoma (*i.e.,* solid and micropapillary) are more likely to have lymph node metastasis (chi-square test, $p = 0.015$), which is consistent with the previous studies (*Kudo et al., 2015*; *Zhang et al., 2021*).

This study has some limitations. First, this is a retrospective study with several unavoidable restrictions, including an inevitable selection bias. Second, due to the lack of multiple comparisons correction in our study, the factors in multivariate regression analysis should have a definite model instead of using meaningful *P* values as predictors. Therefore, future studies with a larger sample size are needed to enhance the generalizability of our findings. Third, the indicators in this study are all man-made observation and measurement, even if the measurement methods have been unified, there is a certain bias, which may cause a certain deviation to the results.

## CONCLUSION

In lung adenocarcinoma with solid density, SUVmax >6.99 combined with HRCT (pleural indentation sign and vascular convergence sign) is helpful to suggest that it is a poorly differentiated group.

## ACKNOWLEDGEMENTS

The author is particularly grateful to the technicians who operated the PET/CT machine.

### Funding

This work was supported by a grant from the Innovation Fund of Harbin Medical University (2020-KYYWF-1446) and the Harbin Medical University Institutional Human Ethics

Review Board approved this retrospective study (No. HMUIRB20160008). The funders had no role in study design, data collection and analysis, decision to publish, or preparation of the manuscript.

## Grant Disclosures

The following grant information was disclosed by the authors:

Innovation Fund of Harbin Medical University: 2020-KYYWF-1446.

Harbin Medical University Institutional Human Ethics Review Board: HMUIRB20160008.

## Competing Interests

The authors declare there are no competing interests.

## Author Contributions

- Xiaolin Chen performed the experiments, analyzed the data, prepared figures and/or tables, authored or reviewed drafts of the article, and approved the final draft.
- Ping Li performed the experiments, analyzed the data, prepared figures and/or tables, and approved the final draft.
- Minghui Zhang performed the experiments, analyzed the data, authored or reviewed drafts of the article, and approved the final draft.
- Xuewei Wang performed the experiments, prepared figures and/or tables, and approved the final draft.
- Dalong Wang conceived and designed the experiments, authored or reviewed drafts of the article, and approved the final draft.

## Ethics

The following information was supplied relating to ethical approvals (*i.e.*, approving body and any reference numbers):

The Harbin Medical University Institutional Human Ethics

Review Board approved this retrospective study (No. HMUIRB20160008)

## Data Availability

The raw measurements are available in the Supplemental File.

## Supplemental Information

Supplemental information for this article can be found online at http://dx.doi.org/10.7717/peerj.15242#supplemental-information.

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
