# Peer review of "Value of preoperative 18F-FDG PET/CT and HRCT in predicting the differentiation degree of lung adenocarcinoma dominated by solid density"

_PeerJ, doi:10.7717/peerj.15242_

## Round 0.1 · original submission · Major Revisions

Please revise the manuscript as the reviewers suggested.

Reviewer 1 ·

Basic reporting

This manuscript is not well organized and needs major amendment. Some figures are not mentioned in the manuscript. The statistical analysis lacks details regarding sample size calculation and data analysis. The interpretation of the Results needs amendment. The English needs improvement by a fluent English speaking expert.

Experimental design

The authors aimed to assess the findings from PET/CT and HRCT to predict the differentiation degree of lung adenocarcinoma. They included the suspected risk factors for analysis using univariate and multivariate logistic regression analysis. The concept is acceptable. However, the details regarding statistical analysis, the interpretation of the Results, and the Discussion have many flaws that need amendment.

Introduction
1. Each abbreviated term should include the full term at the first appearance (SUV in Line 81).
2. The long sentence in lines 97-100 should be rephrased.

Materials & Methods

Statistical Analysis
1. For the study of risk factors using logistic regression analysis, the sample size is crucial to avoid over-fitting of the model. There is no sample size calculation in this study. There are 20 risk factors in Table 3. For risk factors analysis using multivariate analysis, to avoid over-fitting of the model, the sample size should be 10k + 50, where k is the number of risk factors (B. G. Tabachnick and L. S. Fidell, Using Multivariate Statistics, Pearson Allyn & Bacon, Upper Saddle River, NJ, USA, 5th edition, 2007.) The recruited sample number is too small for this study.
2. Line 161: should be “they are expressed as median (interquartile range). The categorical data are expressed as number (%).
3. Which statistics are used to compare the data between groups—please clarify.

Results
1. In Table 2, there are 4 columns to be compared. How did the authors calculate the P-Value? Which statistic did you use? Which column differed with which column?
2. In Table 2, the total number of headings of column 2 should be 60 instead of 68.
3. When I compare the “Vessel convergence” in Table 2 between the Moderate differentiation (30+5) and the Poor differentiation (11+11) using chi-square test, the P value is 0.064; not 0.020 as reported. Please clarify the method of your calculation.
4. As for the “Pleural indentation”, the P value from my calculation is 0.003, not 0.015 in Table 2, please explain your method of calculation.
5. How do you calculate the P value of SUVmax?
6. Please re-calculate all the P values in Table 2.
7. Figure 2 is redundant with SUVmax in Table 2. The values of SUVmax in the manuscript (Lines 188-190) are different from the values in Table 2.
8. The image of Figure 3 does not match the heading and the contents in Line 190.
9. The image of Figure 4 does not match the heading and the contents in Line 195.
10. In Table 4, the authors reported redundant data (columns 2-4 and columns 5-7) with different information. Please clarify.
11. Figure 5 mentioned in Line 218 does not match Figure 5 but matches Figure 8. Please explain and amend.
12. Figures 5-7 have not been referred to in the manuscript.
13. The authors combine 3 risk factors, i.e., vascular convergence, pleural indentation, and SUVmax as a diagnostic tool to predict poorly differentiated lung carcinoma (Line 213-214). The adjusted ORs of vascular convergence and pleural indentation have values of <1 (0.170 and 0.229) which reflect a protective effect while the adjusted OR of SUVmax has a value of >1 (1.402) which shows harm effect. How can the authors combine them? Please explain in detail regarding the method of combination and how to identify the cut-off point.

Discussion
1. In Lines 236-237, the authors state that “pleural indentation and vascular convergence were more likely to occur in poor differentiated”, but the adjusted OR of these two factors has values <1 (0.229 and 0.170) which should be interpreted that they occur less in poor differentiated. Please clarify.
2. The Discussion did not mention details regarding the combination of the 3 risk factors to predict the poorly differentiated tumor.

Conclusions
1. The contents did not give a summary regarding the findings of the study.

Validity of the findings

The finding needs major amendment.

Reviewer 2 ·

Basic reporting

(1) The English is written well.

(2) The title of this manuscript is not appropriate. The authors only mentioned PET/CT, but they also used parameters of high-resolution CT in their model.

(2) Figure 3-6 are images of individual cases. They can be move into supplementary files.

(3) The authors provided sufficient background, relevant results, and references.

Experimental design

(1) Patients with multiple nodules, recurrent or metastatic nodules, history of other tumors or chemoradiotherapy should be excluded. But the authors didn't mention that in their exclusion criteria.

(2) How did the authors distinguish moderate differentiation and poor differentiation? Where are the patients with well differentiation or undifferentiation?

(3) Did The interval time between patients performing CT and undergoing surgery cause bias to the results? The authors should better discuss it.

(4) The authors should better construct a nomogram to visually display their model.

Validity of the findings

(1) The authors should better validate their model using another cohort of patients, or using bootstrapping algorithm.

Additional comments

none

---

## Round 0.2 · Major Revisions

Please revise the manuscript as reviewer 1 has suggested.

Reviewer 1 ·

Basic reporting

There are still many flaws.

Experimental design

There are still many flaws regarding the statistical analyses.

Validity of the findings

Reviewer’s re-review comments

The authors have done hard work revising the manuscript according to the comments. However, there are still major flaws regarding the statistical analyses—the univariate and multivariate analyses.
1. The results of the univariate analysis in Table 3, the authors reported OR of vascular convergence as 0.284 with a 95% CI of 2.095-2.849. The 95%CI range does not include the OR which indicates a wrong calculation.
2. The same problems happen in the results of multivariate analysis in Table 4, the 95%CI range of Pleural indentation and Distortion of air bronchogram do not include the ORs which are incorrect.
3. The ORs of vascular convergence and distortion in Table 3 are less than 1 (0.284 and 0.347), however, both values are more than 1 (2.116 and 2.085) in Table 5. These contradictions reveal the mistake in statistical analyses.
4. Since the results in Tables 3 and 4 are the main outcomes of the study, these flaws
make further analysis based on data from Tables 3 and 4 invalids.
5. If the ORs of pleural indentation, vascular convergence, and distortion have values of <1, they are protective factors for poor differentiation which contradict with the Discussion.
6. The authors should give details on the process of combining different risk factors into one prediction model.

Annotated reviews are not available for download in order to protect the identity of reviewers who chose to remain anonymous.

Reviewer 2 ·

Basic reporting

no comment

Experimental design

no comment

Validity of the findings

no comment

Additional comments

no comment

---

## Round 0.3 · Major Revisions

Please revise the manuscript as the reviewer suggested.

Reviewer 1 ·

Basic reporting

See the following comments.

Experimental design

See the following comments.

Validity of the findings

Reviewer’s comments #3

1. In line 170, the authors stated that “categorical data are expressed as number (%)”, however, the categorical data in Table 1 and Table 2 are expressed as number only—without percentage. Please response.
2. In Abstract lines 36, the authors reported “(mean age 61±8 years) which contradicts with Results line 193, the authors reported “an average age of 60 years (range 37-78 years)”. Please response.
3. In lines 225-227, the authors reported only 95%CI of risk factors, they should report OR together with 95% CI.
4. In lines 230-232, the authors reported the diagnostic efficiency of SUVmax combined with vascular convergence, SUVmax combined with pleural indentation, SUVmax combined with vascular convergence and pleural indentation. Since the values of SUVmax are continuous data while the values of pleural indentation and vascular convergence are categorical data, how did the authors combine them into a new diagnostic tool? For generalization of the results of the study, the authors must clarify in detail the methodology regarding the combination of these risk factors so that the readers can replicate the procedure.
5. In Table 5, the SUVmax has cut-off value more than 1 (6.99) while the SUVmax and vascular convergence, SUVmax and pleural indentation, and SUVmax, pleural indentation and vascular convergence have cut-off points less than 1 (0.31, 0.41, and 0.25) which is quite confusing. The authors should add more details in the Discussion to clarify these aspects so that the readers can understand and be able to replicate the study.

Annotated reviews are not available for download in order to protect the identity of reviewers who chose to remain anonymous.

Reviewer 2 ·

Basic reporting

no comment

Experimental design

no comment

Validity of the findings

no comment

Additional comments

no comment

---

## Round 0.4 · accepted · Accept

This manuscript can be accepted now.

Reviewer 1 ·

Basic reporting

All comments are properly addressed and amended.

Experimental design

All comments are properly addressed and amended.

Validity of the findings

All comments are properly addressed and amended.